# Predicting the Potential Current and Future Distribution of the Endangered Endemic Vascular Plant *Primula boveana* Decne. ex Duby in Egypt

**DOI:** 10.3390/plants9080957

**Published:** 2020-07-29

**Authors:** Mohamed Abdelaal, Mauro Fois, Mohammed A. Dakhil, Gianluigi Bacchetta, Ghada A. El-Sherbeny

**Affiliations:** 1Department of Life and Environmental Sciences, Center for Conservation of Biodiversity (CCB), University of Cagliari, Viale S. Ignazio da Laconi 13, 09123 Cagliari, Italy; mfois@unica.it (M.F.); bacchet@unica.it (G.B.); 2Department of Botany, Faculty of Science, Mansoura University, Mansoura 35516, Egypt; ghada204@mans.edu.eg; 3Department of Botany and Microbiology, Faculty of Science, Helwan University, Cairo 11790, Egypt; Mohamed_dakhil@science.helwan.edu.eg; 4Hortus Botanicus Karalitanus (HBK), University of Cagliari, Viale S. Ignazio da Laconi 9–11, 09123 Cagliari, Italy

**Keywords:** global warming, in situ conservation, population size, Sinaic biogeographic sector, threatened species

## Abstract

Knowledge about population attributes, current geographic distribution, and changes over predicted climate change for many threatened endemic vascular plants is particularly limited in arid mountain environments. *Primula boveana* is one of the rarest and threatened plants worldwide, surviving exclusively in Saint Catherine Protectorate in the Sinaic biogeographic subsector of Egypt. This study aimed to define the current state of *P. boveana* populations, predict its current potential distribution, and use the best-model outputs to guide in field sampling and to forecast its future distribution under two climate change scenarios. The MaxEnt algorithm was used by relating 10 occurrence-points with different environmental predictors (27 bioclimatic, 3 topographic, and 8 edaphic factors). At the current knowledge level, the population size of *P. boveana* consists of 796 individuals, including 137 matures, distributed in only 250 m^2^. The Canonical Correlation Analysis (CCorA) displayed that population attributes (density, cover, size index, and plant vigor) were positively correlated with elevation, precipitation, and pH. Based on the best-fitting model, most predicted suitable central sites (69 km^2^) of *P. boveana* were located in the cool shaded high-elevated middle northern part of St. Catherine. Elevation, precipitation, temperature, and soil pH were the key contributors to *P. boveana* distribution in Egypt. After field trips in suitable predicted sites, we confirmed five extinct localities where *P. boveana* has been previously recorded and no new population was found. The projected map showed an upward range shift through the contraction of sites between 1800 and 2000 m and expansion towards high elevation (above 2000 m) at the southern parts of the St. Catherine area. To conserve *P. boveana*, it is recommended to initiate in situ conservation through reinforcement and reintroduction actions.

## 1. Introduction

Three main challenges face the survival and persistence of rare and threatened endemic vascular plants: Where are they currently distributed? What are the main environmental drivers for their distribution? How can they interact with the predicted future climate changes? Since their level of threat is typically high, endemics are considered as indicators for assessment and conservation of biogeographical regions and biodiversity hotspots [1,2]. Knowledge about the population size, current distribution, and changes over time for many of these species is particularly limited in arid environments. Likewise, field surveys for unknown populations are time-consuming, resource intensive, and most of them are difficult to reach [3,4]. Accordingly, it is crucial to use models that allow more efficient field studies [4]. Model-based sampling is utilized to guide surveys in similar environments where there is a high probability of species presence [3,5].

In mountain areas, global warming is forecasted to affect vascular plants’ diversity [6,7]. In particular, climate changes may result in shifts in the distribution range of rare and threatened endemic species, which might become endangered or even extinct [8]. In order to achieve an effective long-term conservation plan, it is critical to predict the distribution of future climatically suitable areas for rare and threatened species [9].

Species distribution models (SDMs) have increased rapidly in the last two decades and become one of the most commonly used tools in ecology and conservation [10,11]. SDMs operate by correlating a set of known species occurrences with environmental variables to predict where a species is currently found and/or will be found throughout an area of interest [12]. SDMs applications include the study of ecological niche patterns, recognition of suitable localities for conservation and translocation concerns, prediction of future distribution as a result of climate and land-use changes, and to assess fundamental ecological and evolutionary issues [4,13,14]. For unknown populations, the predictive maps of SDMs can help to guide field surveys, search for new populations of poorly known species and better identify spatial distribution areas for monitoring or reintroductions [4,15,16,17,18,19,20]. Conceptualization (e.g., model objective, taxon, location, predictors, scale), data (presence data), model fitting (variables selection, model setting and selection, threshold selection), assessment (performance statistics), and predictions (outputs) are the main steps for building SDMs [21]. SDMs outputs are frequently expressed as the probability of presence or as habitat suitability indexes, ranging from 0 (unsuitable) to 1 (optimal/suitable) [11].

Modelling species with presence-only data has been mostly applied for species with a narrow distribution range and a small number of occurrence records [22]. Maximum entropy (MaxEnt) was preferred among SDMs, due to many reasons. First, the input data are only presence records and its prediction is robust and accurate even with small sample sizes, so that the distribution of endangered species can be predicted well [23]. Furthermore, it provides an explicit spatial map for potential habitats, and allows replicated runs regardless of threshold rules and the contribution percentage of each environmental variable can be computed using the jackknife test [11,23].

In order to obtain meaningful SDMs results, it is necessary to select abiotic variables that relate to the species ecological niche. Different species may have special restrictions related to their reliance on environmental factors and no single variable is assumed to only be important for all species [24]. Climate, topography, edaphic factors, and biotic interactions have been accepted as the main drivers for species distributions at different geographical scales [16,25,26,27,28,29,30,31,32]. As the climate is one of the main driving for species distribution, the bioclimatic variables of the WorldClim database are often used because of its high resolution (~1 km) and quality [33]. In mountains, topography variables (elevation, slope, and aspect) are mostly linked with temperature, wind, UV radiation, and solar radiation quantity; therefore, they play a vital role in the survival and persistence of plant species [7,34]. Soil-related variables are also considered as important agents for the growth and distribution of plant species [24,35].

*Primula boveana* Decne. ex Duby (*P. boveana* hereafter) deserves special attention because it was globally assessed as critically endangered (CR) according to the International Union for Conservation of Nature (IUCN) guidelines [36]. The first step towards the implementation of a conservation strategy for endemic plants is to recognize the geographical distribution, current population status, and threats [37]. To our knowledge, no previous studies have addressed the current and future geographic distribution of *P. boveana*; accordingly, predicting its habitat suitability and unknown populations are critical to conserve or reintroduce this plant species.

In this study, we addressed the following three questions: What is the current state of the population size and range of *P. boveana*? Could SDMs, in particular MaxEnt, guide field sampling to discover unknown populations of *P. boveana*? Finally, will the suitable range of *P. boveana* reduce according to the climate change scenarios? Hence, our objectives were to: (1) Know the current status of *P. boveana* in terms of the size and distribution range; (2) explore the respective power of climate, topography, and edaphic features to predict the current distribution of *P. boveana*; (3) use the best-fit model results to guide field surveys for searching unknown populations; and (4) forecast the potential future distribution (over two time-periods 2050 and 2070) of *P. boveana* under two minimum (RCP2.6) and maximum (RCP8.5) emission scenarios.

## 2. Results

### 2.1. Population Status, Original Habitat Features, and Influencing Variables

The overall known surface area occupied by *P. boveana* is only 250 m^2^. *P. boveana* is distributed in six localities within St. Catherine: Wadi Shaq Mousa (WSM), Wadi Garagenia (WG), Ain Shennarah (AS), Kahf El-Ghoula (KG), Gebal Alahmar (GA), and Sad Abu Hebeik (SH) (Table 1). Six fragmented populations and 10 subpopulations (three each in WSM and WG, and one in each other locality) were recorded. These current locations represent the global population of *P. boveana*. The population size of *P. boveana* consists of 796 individuals, including 137 mature individuals, distributed in the above-mentioned localities. The highest numbers of total and mature individuals were recorded in WSM (547 and 78 individuals, respectively) and WG (176 and 37 individuals, respectively). Similarly, the highest values of plant density, cover, and size index were attained in WSM and WG (Table 1).

*P. boveana* is restricted to a narrow high-elevation range between 1687 and 2208 m above sea level. This species is confined to rock crevices within gorges, slopes, and caves that face NE-E (25.2–70.2°) with a slope above 60° (Table 1). The highest values of plant density and cover were recorded in both gorge and slope microhabitats. The associated species in *P. boveana* populations include, among others, *Scrophularia libanotica* Boiss., *Adiantum capillus-veneris* L., *Hypericum sinaicum* Hochst. ex Boiss., *Origanum syriacum* L., *Mentha longifolia* (L.) L., and *Nepeta septemcrenata* Ehrenb. ex Benth. Our field observations displayed that *P. boveana* is severely threatened by a shortage of rainfall and human activities (human-derived aridification, overcollection for scientific studies, and overgrazing). Occasionally, it is threatened by tourist activities, uprooting by sudden rare floods, and ants attack (local Bedouin observations).

A Pearson correlation matrix (Appendix A) displayed that only the size index is positively correlated with density. On the other hand, along with the Canonical Correlation Analysis (CCorA) (Figure 1), density and cover are positively correlated with elevation, the cover is positively correlated with the precipitation of the driest month (Bio14), while pH and organic carbon are positively related to the size index. Plant vigor is separated in the lower left quarter and positively related to precipitation seasonality (Bio15). Furthermore, bulk density (BD) is negatively correlated with density, cover, and the size index, while the aridity index is negatively correlated with cover.

### 2.2. Models Evaluation and Contributions of Variables

The use of various predictors (climatic, topographic, and edaphic), solely or together, significantly influenced the model’s performance (*p* < 0.05) as measured by the area under the curve (AUC) and true skill statistics (TSS) values. All models except the edaphic-only model indicated high degrees of predictive performances with high values of AUC (>0.90) and TSS (>0.80) (Table 2). The climate-topography-edaphic model showed the best performance as compared with other models (AUC _training_ = 0.993 ± 0.00 and TSS = 0.895 ± 0.02). As expected, the response of each individual predictor was influenced by the different combinations of variables inputted in MaxEnt. For instance, in the climate-only model, the precipitation of the driest month (Bio14), precipitation seasonality (Bio15), and minimum temperature of the coldest month (Bio6) were the highest contributing predictors, while elevation and slope were the most important predictors in the topography-only model (collectively, contributed by 96.4%). For the edaphic-only model, pH, organic carbon, and clay content were the best predictors. The best predictive climate-topography-edaphic model showed elevation, Bio14, Bio15, Bio6, pH, organic carbon, clay, and slope as the eight most important environmental predictors (Table 2). In total, climate variables (contribution of 50.9%) and topographic variables (34.5%) contributed more to the climate-topography-edaphic model than soil variables (14.7%). Of the individual climate, topography, and soil variables, Bio14, elevation, and pH were the most influential for *P. boveana*, respectively. On the other side, the aridity index, aspect, and soil bulk density appeared to be the lowest contributors to the *P. boveana* distribution.

The response curves of the highest eight contributors for the *P. boveana* distribution are shown in Figure 2. Each response curve shows the relationship between each environmental variable and the logistic probability of presence. According to the response curves of topography variables, the most suitable elevations were above 2000 m and no reduction in suitability was found at the highest elevations, whereas suitable slopes were above 35 degrees with an optimum at about 75 degrees. As regards the climatic variables, the optimum minimum temperature of the coldest month (Bio6) was −5 °C, the precipitation of the driest month (Bio14) was above 0.75 mm, and the precipitation seasonality (Bio15) was above 65, and no reduction in suitability by increasing both values of Bio14 and Bio15. Regarding the edaphic predictors, the suitable habitats were attained at a clay content above 11%, soil organic carbon equal to 6 g/kg, and pH value of 7.4. By increasing the values of Bio6 and pH, the habitat suitability of *P. boveana* decreases (Figure 2).

### 2.3. Predictive Potential Current Habitat Suitability of P. boveana

The potential distribution maps of *P. boveana* are displayed in Figure 3A–G. Before refining the models’ outputs, we rejected three edaphic-based models (edaphic-only model, climate-edaphic model, and topography-edaphic model) due to a high false-positive rate (Figure 3A–G, Table 2). We therefore kept the following four models with acceptable performances: climate-only model, topography-only model, climate-topography model, and climate-topography-edaphic model.

According to the climate-only model, out of the 5196 km^2^ of the St. Catherine area, 4816 km^2^ (<0.20) was unsuitable for *P. boveana*, 201 km^2^ with a low habitat suitability (0.20–0.40), 106 km^2^ with moderate suitability (0.40–0.60), and only 73 km^2^ with the highest suitability (>0.60). Similar to the climate-only model, the other three acceptable models (topography-only model, climate-topography model, and climate-topography-edaphic model) predicted the majority of central suitable habitats of *P. boveana* in the high-elevated middle northern sector of St. Catherine. By considering the best-fitting model (climate-topography-edaphic model), the total predicted area of the current highly suitable habitats for *P. boveana* is 69 km^2^, representing 1.33% of the total St. Catherine protected area. For other predictions from different models, see Table 2 and Figure 3A–G.

### 2.4. Potential Areas for New Population Survey or Reintroduction

The cumulative potential areas to survey for unknown populations of *P. boveana* at two thresholds, the maximum training sensitivity plus the specificity and lowest presence threshold (LPT), are shown in Figure 3H. With more than 60% of the probability of presence, the survey areas were lowered to 63 grid cells (63 km^2^). With the exception of current localities, the model predicted as suitable Gebal Catherine, Gebal Mousa, Gebal Safsafa, Gebel Umm Shaumer, Wadi Eltalaa, Abu Tweita, Shaq Itlah, and Elgalt Elazrak. After several field surveys (April to July 2019) in the predicted sites, specifically inside the microhabitats of *P. boveana* (gorges, slopes, and caves) near water springs, we unfortunately did not find any new population. Nonetheless, the models’ outputs confirmed the five extinct localities, which were previously mentioned in the literature (Gebal Catherine, Gebal Mousa, Gebal Safsafa, Gebel Umm Shaumer, and Elgalt Elazrak). In addition to the current localities, the predicted sites could be suitable for translocation activities. Wadi Shaq Mousa was regarded as the best site for the translocation of *P. boveana* according to its high current suitability, which is predicted to persist in the future.

### 2.5. Impact of Climate Change Scenarios on the Future Distribution of P. boveana

The binary future potential distribution area (0: unsuitable/1: suitable) of *P. boveana* for 2050 and 2070 by two representative concentration pathways (minimum RCP2.6 and maximum RCP8.5) are compared with the current climatically suitable areas (climate-only model) in Figure 4. All the predicted suitable sites in the future were included in the range of the study area. In detail, sites of >2500 m elevation that were not included in the training data but present in the study area were not excluded from the predicted future distributions and considered with a suitability value of 0.90. The multivariate environmental similarity surfaces (MESS) function in MaxEnt outputs displayed that the areas occupied by *P. boveana* may lack novel climatic conditions in the future.

However, the predicted climatically suitable areas (73 km^2^) for *P. boveana* by 2050 will decrease by 9.6% (loss = 7 km^2^) and 12.33% (loss = 9 km^2^) under the two representative concentration pathways (RCP 2.6 and 8.5, respectively). A similar trend is observed for 2070 by loss percentages of 16.43% (12 km^2^) and 24.66% (18 km^2^), respectively. The projected climatic map showed a gradual upward range shift with a contraction in the northern parts at sites between 1800 and 2000 m and a range expansion towards high-elevation sites (above 2000 m) in the southern parts of the St. Catherine area.

## 3. Discussion

### 3.1. Limitations of the Study and the Best Set of Predictor Variables

Both the current and future distribution models for *P. boveana* showed a very high accuracy and predictive capacity, supported by both values of AUC and TSS [19]. However, it is necessary to conduct a full assessment of the models’ fitness and understand the limitations of these models to avoid misapplication of their outputs and avoid errors when prioritizing habitats and conservation design [38]. Two of the main limitations of SDMs are the choice of algorithms/methods and predictor variables. In our study, we applied the MaxEnt algorithm, which outperformed other methods with a high accuracy, particularly in the case of rare species [38,39,40]. Moreover, other modelling methods require a large number of occurrence points and some require true absence data, which are often unavailable and assumed to be problematic for low-dispersal rare species [41,42].

In this study, we took advantage of using regional edaphic data as predictors, together with climatic and topographic variables. Although many studies recommended the inclusion of soil factors when modelling plant distributions at local scales [24,35,43,44,45,46,47], it is preferable to use combinations of variables in order to obtain better predictions [48]. Otherwise, the edaphic factors alone are not enough to model habitat suitability for *P. boveana*, but they can help to fine-tune the model that captures the strong influences of climate and topography. In this study, the use of the edaphic-only model produced an unrealistic predictive map, where the suitable habitats are incompatible with the defined habitats and distribution range of *P. boveana* in the St. Catherine area. These results are comparable to Hageer, Esperón-Rodríguez [49], who applied MaxEnt to model the distribution of 29 Australian plants and emphasized the significant importance of climatic variables over soil variables. Some ecological explanations may be missed because of the lack of a large set of variables at a higher resolution. Indeed, the use of high-resolution soil predictors with topo-climatic predictors improved, in some cases, the predictive power of plant SDMs [50,51,52].

### 3.2. The Predicted Current Suitable Sites for Survey or Translocation of P. boveana

Our results demonstrated that the current habitat suitability of *P. boveana* extends within the middle northern boundaries of the St. Catherine area. Based on the LPT threshold, the potential distribution areas for the search for unknown populations of *P. boveana* are minimized, rendering the efforts feasible. After several field surveys during the flowering seasons (April–July), we did not find any new population, but we confirmed five extinct localities/populations, namely Gebal Catherine (2113 m), Gebal Mousa (2285 m), Gebal Safsafa (2166 m), Gebel Umm Shaumer (2090 m), and Elgalt Elazrak (2150 m). This finding fits with the known historical distribution reported in previous literature [53,54,55,56,57,58,59,60,61]. In detail, Danin (1983) [53] indicated that *P. boveana* has been found in Gebal Catherine, Gebal Safsafa, and Gebel Umm Shaumer while St. Catherine rangers also reported its presence in both Gebal Catherine and Elgalt Elazrak between 2007 and 2012, but then it completely disappeared. Such local extinction in suitable sites may be attributed to drought, habitat fragmentation, and human activities (aridification by the collection of water for consumption, sheep and goat grazing, and overcollection) [56,58]. However, additional annual field samplings are recommended in suitable sites for searching new *P. boveana* populations or to examine the factors that shared in blocking the colonization and recovery of this plant in all suitable historic sites.

All of the current predicted sites satisfy the *P. boveana* growth conditions of a high elevation (~>2000), slope > 35 degrees, with an optimum precipitation seasonality of ~75 and not less than ~65. Consequently, warm and dry sites with an elevation < 2000 m are less suitable for *P. boveana*. These results are in complete accordance with [56,59,61], who reported that *P. boveana* largely occurs in moist, shaded, and north-facing rock crevices at low temperature with an elevation range from 1800 up to 2210 m.

### 3.3. Main Environmental Predictors for the Distribution of P. boveana

The CCorA ordination and variable’s contribution percent in MaxEnt indicated that the *P. boveana* distribution was more sensitive to topographic variables (elevation and slope), precipitation (Bio14 and Bio15), temperature (Bio6), and soil factors (pH, organic carbon, and clay). The best population status of *P. boveana* is at high elevation and in alkaline and moist soils. Our results are congruent with other studies that dealt with rare, endangered, and medicinal mountain plants, where they addressed the crucial role of low temperature, high precipitation, and elevation in plant distribution and fitness. For instance, *Artemisia sieberi* Besser and *A. aucheri* Boiss. [52], *Primula scandinavica* Brunn [7], and *Daphne mucronata* Royle [62]. Specifically, in the study area, climate and elevation are the main contributing environmental variables for the distribution of endemic taxa, in particular, *Rosa arabica* Crép. [9], *Nepeta septemcrenata*, and *Hypericum sinaicum* [63,64]. Moreover, the elevation is regarded as the main driver for the survival and persistence of mountain plant species [7,65,66], but also to some degree, slope and aspect have played important roles in the microclimate of narrow-range species [67]. The elevation is often connected with changes in temperature and solar radiation, which might affect plant growth [34]. Among soil factors, pH is considered the main predictor for plant distributions and significantly improved the predictive capacity of SDMs [35,49,50,68]. Soil pH is frequently associated with nutrient availability [69].

All previously published literature confirmed a continuous decline in the habitat quality of *P. boveana*, with evidence of a decline in size and number with time [54,56,58,59,61]. In detail, the population size has fluctuated as follows: 2000 individuals in 1991 [54], 336 individuals in 2007, 268 individuals in 2011, 115 individuals in 2013 [58], 1010 individuals in 2014 [59,60], and 796 individuals in the current study. The fluctuating population size, especially between 2007 and 2013, may be more probably due to a sampling and/or detectability artefact, because perennial rocky species, such as *P. boveana*, uncommonly show such a high annual fluctuation. In contrast, the trend highlighted by field surveys conducted in 1991, 2014, and 2018 (current study) seems to really be a sign of a continuous decline. The alarming reduction in the *P. boveana* population is apparently related to the growing aridification (natural or human-derived); human activities; rare gene flow between nuclei; deep seed dormancy; high level of inbreeding, which frequently causes a drop in fitness; limited seed dispersal; habitat fragmentation; and an increase in temperature [56,58]. All of these threats may force *P. boveana* toward extinction.

### 3.4. Future Predictive Distribution Area of P. boveana under Two Global Warming Scenarios

Habitat suitability decline and upward range shifts due to future global warming for the years 2050 and 2070 are predicted for *P. boveana*. This is in agreement with Hoyle and James [70], who expected such a decline for all species endemic to the Sinai mountains. Furthermore, a recent study in the same area confirmed this pattern of range shifts for the endemic plant *Rosa arabica* [9]. The mountaintop and range-restricted species will respond to the predicted global warming by shifting their range boundaries towards high elevations [71]. Subsequently, predicted global warming might adversely affect endemic species that currently survive at the highest elevation of St. Catherine mountains [15]. Less annual precipitation would inevitably reduce water availability and moist soils, therefore lowering the quality and size of *P. boveana*-suitable habitats. Moreover, increasing human disturbance on St. Catherine mountains would exacerbate the water availability problems, which affects the survival of this plant [71]. Following previous studies [57,58,59], our field observations displayed that grazing is a key pressure on *P. boveana* and this might explain why this species is often found on very steep cliffs. In mountainous environments, site properties vary greatly at a small scale due to differences in elevation and aspect, thus *P. boveana* was at a north-facing site and hence it annually receives little solar radiation. Therefore, its current presence inside shaded and cool microhabitats (gorges, slopes, caves) may represent refuges for future conservation against the expected increases in temperature. Finally, in future projections, uncertainty may increase, especially when the environmental predictors (such as elevation in our study) need to be extrapolated outside the range of the training data of the species’ response; therefore, physiological, biological, and distributional attributes of the species should be considered [72].

## 4. Materials and Methods

### 4.1. Study Area and Species

Our study was conducted in Saint Catherine Protectorate (St. Catherine) in the Sinaico-Arabian biogeographic sector and, more specifically, in the Sinaic subsector [73]. It is located in the northeastern corner of Egypt and occupies an area of ca. 5196 km^2^ (Figure 5A,B). The Sinaic subsector supports remarkable biodiversity, with a high percentage of endemic and rare vascular plants, distributed in a wide range of mountain microhabitats (e.g., slopes, gorges, cliffs, terraces, and caves) [67,74]. The Sinaic subsector is a smooth-faced outcropping igneous massif, with an elevation up to 2640 m a.s.l [75]. The rainfall is scarce, intermittent, and reaches monthly averages of 37.5 mm (October–May, 1970–2017), even though unpredictable one-day flash floods reaching c. 300 mm recently occurred (2012–2014) [76]. The average monthly temperature ranges from 8.6 °C in January to 25.5 °C in August. The Sinaic biogeographic subsector hosts 14 endemic vascular plants, about a quarter of the endemic plants to Egypt, so it is considered the most important micro-hotspot in Egypt [9,73,77]. The most important threats to biodiversity are drought and human activities [78].

The Sinai primrose, *Primula boveana* (Primulaceae), is the only species included within the genus *Primula* in the flora of Egypt (Figure 5c). *P. boveana* is a glabrous rhizomatous perennial herb up to 40 cm in height with an erect unbranched stem, sessile leaves, and capsule-type fruit with dust seeds [79]. Globally, *P. boveana* is one of the rarest and critically endangered vascular plants [58,59,80]. It is exclusive to the St. Catherine Protectorate in the Sinaic biogeographic subsector of Egypt [53,81]. As most of the mountainous species of this genus [82], this plant generally grows in cliffs near water-springs, where the interstitial soils are rich in moisture content and organic matter [55,56,60]. The vegetative growth of *P. boveana* blooms over the four seasons with maximum activity in September [56]. For the long-term conservation of *P. boveana*, two managed enclosures (Kahf El-Ghoula and Gebal Alahmar) were established [56].

In this study, the current georeferenced occurrence points of *P. boveana* were obtained from field data (from April 2016 to September 2018) using a Global Positioning System (GPS; Garmin e-Trex 20) and after consulting all the previous literature [36,53,54,55,56,57,58,59,60,80], local Bedouins, and experts’ knowledge. All the currently known localities with the species were visited and 10 permanent plots of 5 × 5 m each were centrally positioned to measure the following parameters: population size (estimated number of individuals in each subpopulation), plant density (number of individuals per plot area), size index (average of heights and diameters (cm) of three randomly individuals per plot), plant vigor (the ratio between the plant size and number of leaves), and cover percent (visually estimated) [83]. Moreover, the associated species and threats were recorded in each subpopulation.

In order to reduce spatial autocorrelation and the associated inflation of model performance, *P. boveana* occurrence records were filtered by excluding redundant occurrence points in each 1 × 1 km grid. Occurrence records were assessed in the ArcGIS 9.3 environment to eliminate spatially correlated points [13,14]. Consequently, 10 occurrence points of *P. boveana* were used to create SDMs (Figure 5b). These occurrence points represent the total known occurrence of *P. boveana* in Egypt.

### 4.2. Environmental Predictors

Three sets of environmental variables were used for predicting the potential suitable distribution of *P. boveana*: bioclimatic variables (27), topography (3), and edaphic factors (8) (Appendix A). Out of 27 climatic variables, 19 variables for the current period (1950 to 2000) were downloaded from the WorldClim v.2 database (http://www.worldclim.org) [84] at a resolution of 30 arc-seconds (~1 km^2^), while the remaining eight variables were obtained from the ENVIREM dataset v.1.0 (http://envirem.github.io) [85] at the same resolution. These data have been widely used to generate species distribution models, as they are relevant to the ecology and physiological response of different plant species [9,14,73]. Elevation was downloaded from the DIVA-GIS online database (https://www.diva-gis.org/gdata) at a 90-m resolution, while slope and aspect were extracted from elevation data in the ArcGIS and then resampled into a 1-km spatial resolution [86]. Edaphic factors related to physical and chemical soil properties at depth intervals of 0–0.30 m were obtained from the SoilGrids database v.0.5.3, available from ISRIC-World Soil Information [87] at the same resolution (1 km^2^) (Appendix A). All of these predictors were a continuous type.

To avoid collinearity problems among variables, the variance inflation factor (VIF) of 38 environmental variables was tested using the ’sdm’ package [88] in R-environment [89]. We performed a stepwise variable selection: First, VIF values for all variables were calculated, then the variable with the highest VIF was iteratively removed until no variables with VIF greater than the threshold (5) remained [9,14]. As a result, 12 variables were kept to develop the models (Table 3). These variables comprised five bioclimatic (Bio6: minimum temperature of the coldest month, Bio7: temperature annual range, Bio14: precipitation of the driest month, Bio15: precipitation seasonality and aridity index), three topographic (elevation, slope, and aspect), and four edaphic factors (bulk density, clay, organic carbon, and pH) (Table 3). Similar to the other mountain species of St. Catherine that are sensitive to high altitude, temperature, rainfall, and soil features [75,78], we assumed that the selected variables were appropriate for defining the ecology and spatial distribution of *P. boveana*.

Prior to the modelling step, a Pearson correlation matrix among population field parameters (density, cover, size index, and plant vigor) was conducted. Moreover, a multivariate Canonical Correlation Analysis (CCorA) was used to investigate the effect of the selected environmental variables on the different population parameters of *P. boveana*.

For future climatic projections, the climatic variables of the Community Climate System Model (CCSM4) over two periods 2050 (average for 2041–2060) and 2070 (average for 2061–2080) under two representative concentration pathways (RCPs 2.6 and RCPs 8.5) of the 5th report of Intergovernmental Panel on Climate Change [90] were downloaded from the WorldClim v.2 (http://www.worldclim.org) [84]. The CCSM4 model has been used broadly in referring to the impacts of climate change on plant distribution [9,91,92,93].

### 4.3. MaxEnt Modeling Procedures

MaxEnt software version 3.4.1, [11] (https://biodiversityinformatics.amnh.org/opensource/maxent/) was utilized to predict the current and future suitable habitats’ distribution of *P. boveana* in Egypt. To predict the potential occurrence of *P. boveana*, we built seven MaxEnt models depending on the selected environmental variables. We used climate variables only, topography variables only, edaphic variables only, and both climate and topography, climate- edaphic variables, topography- edaphic variables, and finally climate, topography, and edaphic variables. We employed 10 replicates, with cross-validation replicates for each model type [9,94]. The relationships between selected variables, and the probability of the presence of *P. boveana* were displayed in MaxEnt’s response curves. The percent contribution of each environmental variable was calculated, and jackknife procedures were executed to assess each variable’s contribution [95]. The remaining MaxEnt settings were set to default values.

To measure the predictive performance of the model, two metrics were used, the area under the curve (AUC) derived from the receiver operating characteristic (ROC) curve [96] and true skill statistics (TSS, threshold value = 0.5) [97]. Higher values of AUC and TSS (closer to 1) indicate higher accuracy and confirm a relationship between model prediction and distribution. The significant difference between values of AUCs and TSS among different models was tested using the Kruskal–Wallis one-way ANOVA test. All descriptive statistics and CCorA were performed using the XLSTAT 2016.

The predictive map with four classes of habitat suitability for *P. boveana* was defined using natural breaks in ArcGIS: Unsuitable (<0.20), low suitability (0.20–0.40), moderate suitability (0.40–0.60), and high suitability (>0.60) [9,98,99]. Differences in the predicted current ecological range of *P. boveana* among the different MaxEnt models in four classes were computed in ArcGIS using selection by attributes and the total number of grid cells in each class was counted and converted into surface areas (km^2^). The selection of the best-fitting model was based on the minimal predicted highly suitable areas [41].

To validate our model and discover unknown populations for *P. boveana*, we carried out a new MaxEnt model using the highest contributed environmental predictors (Bio6, Bio14, Bio15, elevation, slope, clay, organic carbon, and pH). We selected the cumulative outputs with values of relative suitability from 0% to 100%, to preclude assumptions about species prevalence [11]. Two thresholds were considered to generate binary maps of presence/absence. The first threshold was the maximum training sensitivity plus the specificity [100]. The second was the lowest presence threshold (LPT), which was used to identify both the predicted minimum area and unknown distribution areas [4,17,22]. The resulting two maps were superimposed to generate a map of potential areas for field surveys. All predicted sites with a relative suitability of more than 60% were considered as highly potential suitable sites for ground-truthing [17,99]. We made a set of georeferenced field trips from April to July 2019 to the high-suitable sites to search for unknown populations/localities of *P. boveana*.

For future projections, in order to increase the reliability of models, we ran the multivariate environmental similarity surfaces (MESS) analysis in MaxEnt [23]. The MESS function investigates the univariate extrapolation under the different future scenarios, determines novel climatic conditions, and assesses if values of every single variable under a projection scenario are out of its range of values under current conditions. Moreover, to predict distributional changes due to future climate change, we compared distribution changes between current climatically suitable habitats and two future scenarios in the ArcGIS environment. Firstly, we converted the climate-only model map into a binary map (1: suitable/0: unsuitable) by choosing a threshold value of 0.60 using SDM toolbox v.2.4 [101]. Then, the loss or gain in the suitability area was calculated given the difference between the current and future climatically suitable areas [9].

## 5. Conclusions

This study demonstrated that, for more realistic predictions of the distribution of *P. boveana*, it is better to include soil variables, not alone, but together with climate and topographic variables in the modelling procedure. Even if the values of MaxEnt models’ AUC and TSS scores were closely high, it is critical to emphasize the value of ground-truthing validation of model outputs, rather than depending uniquely on common metrics for accuracy assessment. This is especially true when models are constructed from original presence records of the native locations for the species. However, we obtained predicted results consistent with their actual distribution. Additionally, the models provide valid information on the ecological preferences of the taxa. In particular, this study showed that the geographic distribution of *P. boveana* might undergo an upward range shift via a contraction of sites between 1800 and 2000 m and expansion towards high-elevation sites (above 2000 m) at the southern parts of St. Catherine. *P. boveana* distribution was more sensitive to topographic variables (elevation), precipitation (Bio14 and Bio15), temperature (Bio6), and soil pH. To overcome the predicted fluctuation or extinction in the population size of *P. boveana*, it is recommended to initiate in situ conservation through translocations (reinforcements or reintroductions) or the establishment of suitable managed fenced enclosures, and further, it is important to protect it against grazing. Even if some efforts were already overtaken in this sense, our results might support further future conservation plans. For instance, Wadi Shaq Mousa was found to be the best site for the translocation of *P. boveana*, since it has the most suitable current conditions, which were predicted to persist in the future.

## Figures and Tables

**Figure 1 plants-09-00957-f001:**
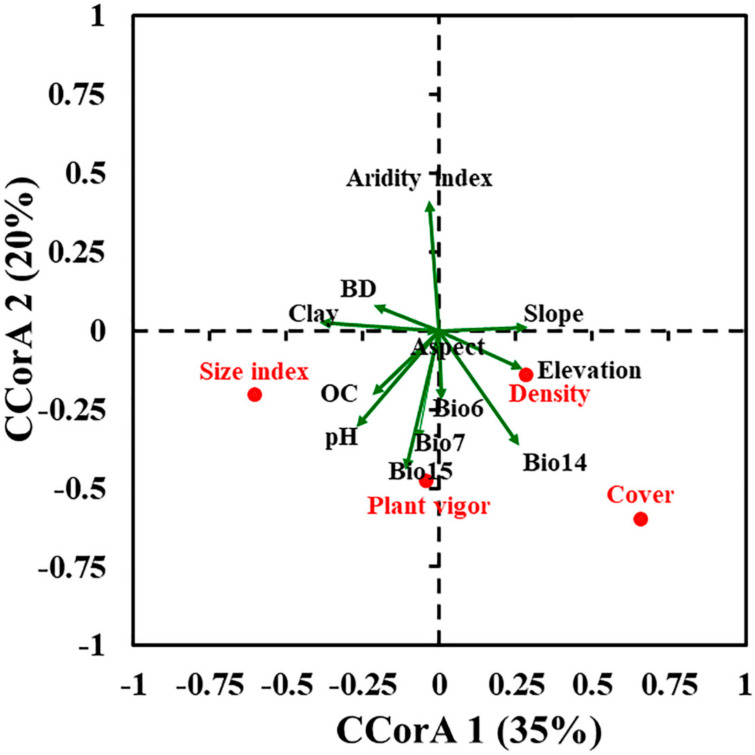
Canonical Correlation Analysis (CCorA) ordination plot between field population parameters (density, cover, size index, and plant vigor) and environmental variables. Variance percentages are indicated after the axes. Environmental variables include minimum temperature of coldest month (Bio6), temperature annual range (Bio7), precipitation of driest month (Bio14), precipitation seasonality (Bio15), organic carbon (OC), and bulk density (BD).

**Figure 2 plants-09-00957-f002:**
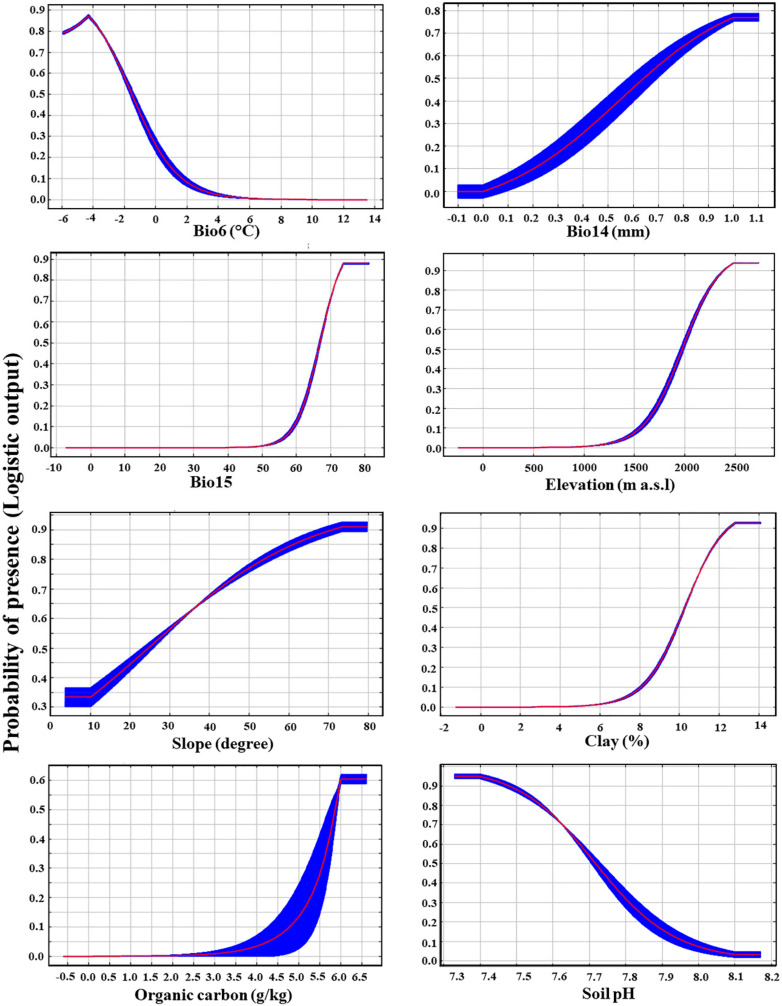
Response curves for the highest contributed predictors in *Primula boveana* current distribution models. The red curves show the mean response of the 10 replicated runs and the blue shades represent +/− one standard deviation. Y-axis values are the predicted probability of habitat suitability, as provided by the logistic output format.

**Figure 3 plants-09-00957-f003:**
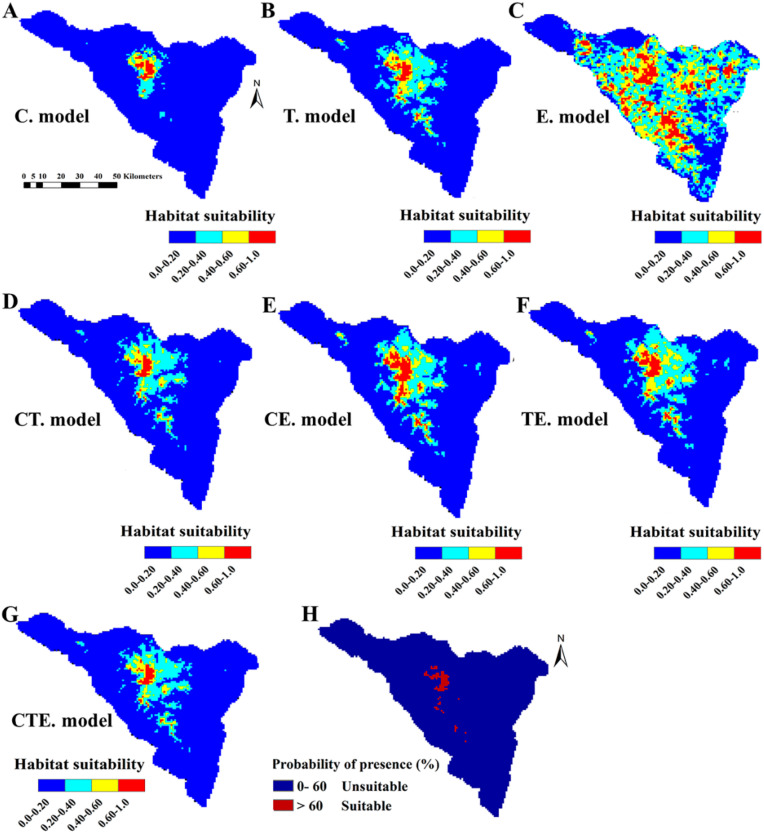
Predictive map for the current habitat suitability (**A**–**G**) of *P. boveana* using seven sets of predictor variables (climate only: C. model, topography only: T. model, edaphic only: E. model, climate-topography: CT. model, climate-edaphic: CE. model, topography-edaphic: TE. model, climate-topography-edaphic: CTE. model) in the St. Catherine area; habitat suitability classes include unsuitable (0–0.20), low suitability (0.20–0.40), moderate suitability (0.40–0.60), and high suitability (0.60–1.0); and (**H**) potential area probabilities for the survey of unknown, new, or historic populations of *P. boveana*. Cumulative percentage according to the lowest presence threshold (LPT) and restrictive threshold of 60% of the probability of presence. 0–60% unsuitable and >60% suitable.

**Figure 4 plants-09-00957-f004:**
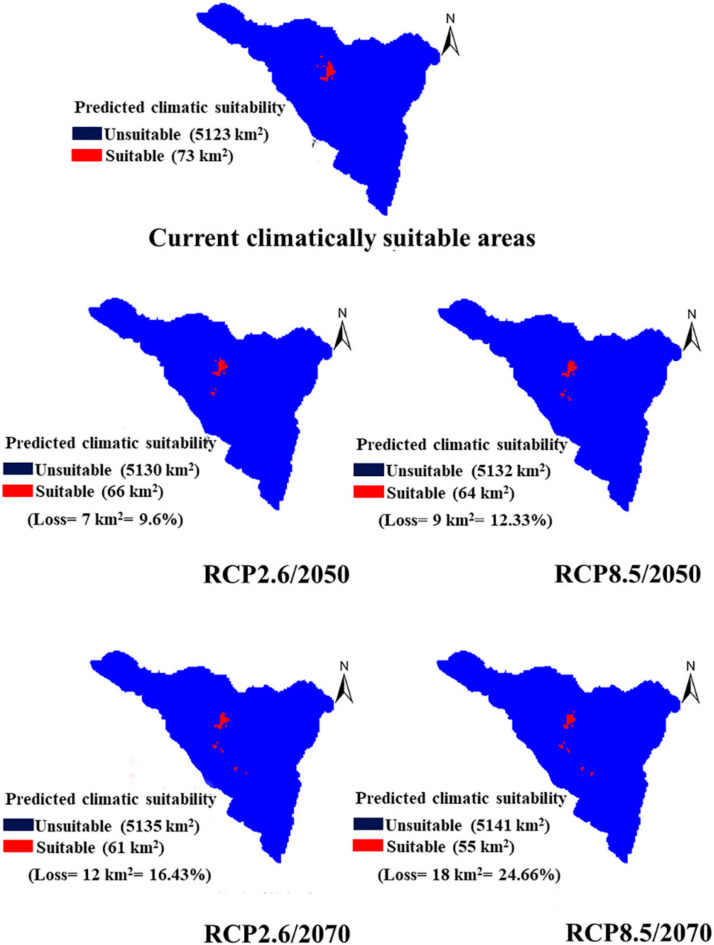
Future potential distribution area of *P. boveana* predicted for 2050 and 2070 under two representative concentration pathways (RCP2.6 and RCP8.5), compared with the current climatically suitable areas (binary predictive map of the climate-only model). The values in the figures represent the area (in km^2^) of the suitable and unsuitable climatic areas, and the loss range (in both km^2^ and percentage).

**Figure 5 plants-09-00957-f005:**
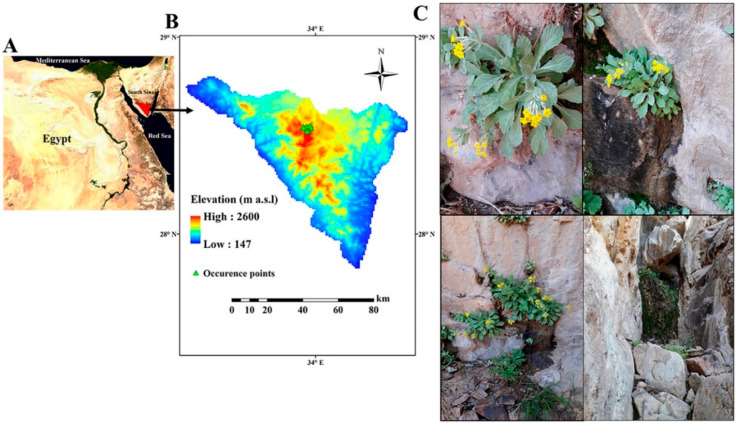
(**A**) Map of Egypt showing the location of St. Catherine Protectorate (red shaded) inside the Sinaic biogeographic subsector. (**B**) Current occurrence points of *Primula boveana* and (**C**) habitat of *Primula boveana*.

**Table 1 plants-09-00957-t001:** The known current locations, population field parameters (mean ± SD), topography, and microhabitat of *Primula boveana* in Egypt.

Parameter		Location									
		Wadi Shaq Mousa (WSM)			Wadi Garagniah (WG)			Maen Shennarah (MS)	Kahf El-Ghoula (KG)	Gebal Alahmar (GA)	Sad Abu Hebeik (SH)
Population parameters											
Total individuals		408.3 ± 12.6	77.7 ± 1.5	61.3 ± 3.1	74.7 ± 0.6	91.6 ± 1.5	10.0 ± 1.0	54.0 ± 1.0	4.0 ± 0.0	7.0 ± 0.0	5.0 ± 0.0
Mature individuals		55.0 ± 2.0	15.0 ± 2.0	8.0 ± 1.0	11.3 ± 1.5	22.7 ± 1.5	2.7 ± 0.6	14.0 ± 1.0	0.7 ± 0.6	5.0 ± 1.0	0.7 ± 0.6
Density (individuals/25 m^2^)		16.8 ± 1.3	3.1 ± 0.3	2.5 ± 0.1	3.0 ± 0.5	3.7 ± 0.8	0.4 ± 0.0	2.2 ± 0.2	0.2 ± 0.0	0.3 ± 0.0	0.2 ± 0.0
Cover (%)		45.0 ± 1.5	25.0 ± 0.5	26.7 ± 2.5	25.0 ± 3.0	35.0 ± 5.0	5.0 ± 0.9	25.0 ± 1.0	2.0 ± 0.0	5.0 ± 0.0	1.0 ± 0.0
Size index (cm)		22.7 ± 2.1	12.0 ± 2.6	16.0 ± 3.0	12.3 ± 1.5	10.0 ± 2.6	14.3 ± 2.0	16.3 ± 1.5	12.0 ± 2.0	10.0 ± 2.0	12.0 ± 2.6
Plant vigor		1.1 ± 0.2	1.2 ± 0.6	1.4 ± 0.0	1.6 ± 0.2	2.7 ± 0.5	1.1 ± 0.0	1.3 ± 0.0	1.0 ± 0.0	1.6 ± 0.2	1.0 ± 0.1
Topography	Elevation (m)	2065	2050	1950	2165	2208	1890	2032	1803	1915	1687
	Slope (degree)	90	90	85	70	90	60	90	90	80	90
	Aspect (degree, direction)	40.3 (NE)	25.2 (NE)	30.3 (NE)	33.2 (NE)	25.5 (NE)	30.3 (NE)	45.5 (NE)	69.7 (E)	70.2 (E)	27.9 (NE)
Microhabitat		gorge	gorge	gorge	gorge	slope	gorge	gorge	cave	slope	gorge

**Table 2 plants-09-00957-t002:** Area under the curve (AUC) and true skill statistics (TSS) of models’ performance (±SD), average percent contribution of the different predictors, and predicted habitat suitability class (in km^2^) for *P. boveana* under different MaxEnt models.

	Climate-Only Model	Topography-Only Model	Edaphic-Only Model	Climate-Topography Model	Climate-Edaphic Model	Topography-Edaphic Model	Climate-Topography-Edaphic Model
Model performance							
AUC training	0.991 ^bcd^ ± 0.00	0.990 ^abc^ ±0.00	0.845 ^a^ ± 0.01	0.992 ^cd^ ± 0.00	0.990 ^cd^ ± 0.00	0.989 ^ab^ ± 0.00	0.993 ^d^ ± 0.00
AUC test	0.989 ^b^ ± 0.01	0.990 ^b^ ± 0.01	0.833 ^a^ ± 0.07	0.991 ^b^ ± 0.01	0.985 ^ab^ ± 0.01	0.987 ^ab^ ± 0.01	0.990 ^b^ ± 0.01
TSS	0.889 ^cd^ ± 0.07	0.884 ^abc^ ± 0.02	0.725 ^a^ ± 0.30	0.893 ^b^ ± 0.05	0.887 ^ab^ ± 0.09	0.882 ^bc^ ± 0.20	0.895 ^d^ ± 0.02
Average percent contribution							
Bio6 (°C)	25.2			15.1	12.8		15.2
Bio7 (°C)	3.8			0.6	1.4		1
Bio14 (mm)	40.4			24.5	31.7		18.1
Bio15	28.6			17.5	25.1		16.4
Aridity index	2			0	1.8		0.2
Elev. (m)		80.3		38.3		68.4	30.4
Slope (degree)		16.1		4		3	4
Aspect (degree)		3.6				0	0.1
BD (g/cm^3^)			2.4		0.2	0	0.6
Clay (%)			22		1.3	6.2	3.5
OC (g/kg)			36.2		10.1	7.3	4.6
pH			39.4		15.5	13	6
Predicted habitat suitability class							
<0.20	4816	4158	1804	4210	4000	2697	4303
0.20–0.40	201	698	1980	679	722	813	632
0.40–0.60	106	262	1090	225	296	332	192
>0.60	73	78	322	82	178	159	69

Different letters in the same raw mean significant difference at *p* < 0.05. Predicted habitat suitability classes include unsuitable (<0.20), low suitability (0.20–0.40), moderate suitability (0.40–0.60), and high suitability (>0.60).

**Table 3 plants-09-00957-t003:** The environmental predictors used in the habitat suitability modeling for *P. boveana* with their variance inflation factors (VIFs < 5). Units were reported for dimensional variables.

Category	Code/Unit	Predictors	VIF	Source and Resolution
Bioclimatic	Bio6 (°C)	Min temperature of coldest month	4.80	WorldClim v.2 (~1 km^2^).
	Bio7 (°C)	Temperature annual range (Bio5-Bio6)	2.73	
	Bio14 (mm)	Precipitation of driest month	2.30	
	Bio15 (unitless)	Precipitation seasonality (coefficient of variation)	4.23	
	Aridity index (unitless)	Degree of water deficit below water need	3.65	ENVIREM (~1 km^2^).
Topographic	Elev (m. a.s.l)	Elevation	3.62	DIVA-GIS (90 m)
	Slope (degree)	Slope	1.58	Derived from Elev.
	Aspect (degree)	Aspect	4.17	Derived from Elev.
Edaphic	BD (g/cm^3^)	Bulk density	3.18	SoilGrids (1 km^2^)
	Clay (%)	Clay content	3.44	
	OC (g/kg)	Organic carbon content	4.50	
	pH	pH in H_2_O	1.38	

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
