# Peer review of "Predicting the Potential Current and Future Distribution of the Endangered Endemic Vascular Plant Primula boveana Decne. ex Duby in Egypt"

_plants, 2020, doi:10.3390/plants9080957_

Round 1
Reviewer 1 Report
This manuscript uses the MaxEnt species distribution model (SDM) with the present-day occurrences of the focal species, based on thorough field surveys (at least, I guess they were thorough; I found no information about these field surveys except their dates in L365), to model habitat suitability for that species, both now and under future climate scenarios. The authors conclude that the species is currently restricted to a few small sites in a small area of Sinai, that the total suitable habitat area is small (~70km2), that this suitable area is likely to decrease with climate change, and that other previously known populations of the species are now extinct. The focal species is critically endangered and of considerable conservation interest, making this a worthwhile study.
SDMs can be very powerful tools for interpolative purposes, such as finding suitable sites for species, as done here, and indeed this is an approach that has worked well in the Sinai Protectorate for other species. When using SDMs for predicting future distributions under climate change, many assumptions are made about transferability, etc, and the results are much less reliable; even so, the approach is commonly used and may have some value if the caveats are taken properly into account. As far as I can tell, the modelling has been done quite well and many of the caveats have been taken into account. However, the explanation of the methods can improve and I have some concerns about some of the specifics of those methods (based in part on having to guess). Overall, I think the results should be reasonably robust to my concerns about the methods, and are unlikely to change qualitatively, but they may change a bit quantitatively.
My main concern is about whether there is any circularity in the results for reduction in suitable area with climate change. I am not sure about this because the method has not been well enough explained. Do the predicted suitable sites in the future (Figure 4) include areas outside the range of the data in the models? For example, the study area includes elevations up to 2640m (L344) but there is no representation in the training data of sites above 2500m (Fig.2). Yet Fig.2 suggests that an elevation of ~2600m would be highly suitable. If such areas were excluded (as implied by L285) then the result that a loss of suitable area for the species is expected in the future is partly circular. If there is a problem here, I do not expect it to qualitatively change the finding (I would still expect reductions in the suitable habitat), but it could change it quantitatively (the predicted amount of reduction).
The procedure to reduce collinearity problems, as described in L392-98, is also problematic. The VIF basically measures the overlap of explanatory power between the focal variable and the other variables in the data. It is good for highlighting where problems are, but it is not appropriate to just remove all variables with high VIF, which is what was apparently done. A simple example illustrates the problem: if you have two variables that measure the same thing and are highly co-linear then both would have a high VIF; these variables might for example be mean temperature and maximum temperature. If those were the only two temperature variables in the data then removing them both would remove temperature completely from the analysis, which is likely to remove an important signal. Given the strong overlap in explanatory power, the correct procedure would be to remove one of the two temperature variables and then re-examine the VIF - in this case, the VIF of the other one would then become very low. The decision on which of the two to remove is best done using theoretical reasoning (which variable is the most appropriate to keep?) Do NOT remove both variables just because they both have high VIF! It looks as though the authors have made this mistake. If so, they should check whether their analysis should be redone. The good news is that I do not expect any such new analysis to alter the results very much. This is because the variables the authors did include seem to me to have been appropriate ones. Perhaps that was good luck! Or perhaps it was actually due to the authors using their judgment about which variables are appropriate to include, and not just about removing all variables with high VIF - in which case the results are fine but the authors should rewrite that part of the methods to explain what they actually did!
My other main comment is about the likely causes of local extinctions and the overall endangerment of the species. L278-80 and section 3.4 cover some issues of clear relevance, but do not mention grazing. Surely grazing is a key pressure on the species, too? My guess is that it is very sensitive to grazing (perhaps by goats?), which might explain why it is most often found on very steep cliffs! If grazing is indeed important then an important part of trying to conserve the species would presumably be to protect it from grazing as much as possible.
SPECIFIC AND/OR MINOR ISSUES in no particular order:
Throughout the manuscript, attention is needed to the English. However, in most cases the meaning is reasonably clear.
In the abstract, ‘we confirmed five extinct localities’ (L32) is not very clear. This eventually becomes somewhat clear (though not very), later in the manuscript, but it should be better explained in the abstract.
L386 states that the elevation data were 1km2 resolution, and from these slope and aspect were calculated. Surely a 1km2 raster digital elevation model is much too coarse to calculate slope and aspect! (Thus I wondered, on reading this, why you did not use the SRTM digital elevation model, available at 90m and 30m resolution.) Yet the slope values for Table 1 suggest that the slope and aspect cannot have been calculated with 1km2-resolution elevation data. So I suspect there is a mistake in explaining the dataset and/or calculation procedure.
Figure 1 and Table S1: given the strong correlation between size and density, why does Figure 1 show them as apparently uncorrelated? This is not necessarily an error, but requires some explanation.
In Table 2, you need to explain in the caption what the various models are (e.g. E. model = edaphic only), as done in Figure 3. Readability is seriously reduced if the reader has to search the manuscript to interpret this table.
Similarly, in the main text, it would be better not to use these abbreviations. It would be quite succinct enough, and much more readable, to refer to the models as, for example, ‘edaphic-only model’, ‘climate-topography model’, etc.
In Figure 2, it would be better for the vertical axes to all have the same scale.
In reporting the results illustrated in Fig.2, the authors should bear in mind that the curve does not peak in most cases, but instead has its maximum value at one end of the data range. Therefore it is not appropriate to make statements such as ‘suitable elevation range was between 1600 m up to 2500 m. a.s.l.’. In this case it would be much more appropriate to say that the most suitable elevations were the highest ones sampled, or at least to point out that no reduction in suitability was found at the highest elevations. Similar considerations apply to most of the other statements about the findings shown in Fig.2.
It is also strange that the statements about ‘suitable range’ encompass almost all non-zero suitability values. Taking the example of elevation again, 1600m is specified as ‘suitable’ but the suitability shown on Fig.2 for this elevation is only 0.1, which contradicts the suitability statements later in the text (e.g. L181), and Fig.3H, where ‘suitable’ is defined as >0.60. Again, similar considerations also apply to the other statements in the text about the results shown in Fig.2.
Section 3.1 in the discussion includes a lot of text about the soil variables, when really the message seems rather simple: that edaphic factors alone are not enough to model habitat suitability for this species, but they can help fine-tune a model that captures the stronger influences of climate and topography. This lengthy discussion of soil influences seems quite pointless because surely no-one would think it sensible to ONLY include soil variables in SDMs!
L436 ‘The selection of the best overfitting model’. Very strange statement; surely you do not mean this! Should it be ‘The selection of the best-fitting model’?
I wonder whether the sites referred to as ‘Wadi Shag Mousa’ (L216, 469) and ‘Wadi Shaq Mousa’ (L109, Table 1) are actually the same site?
Author Response
Dear Reviewer,
We really appreciate your constructive valuable comments and suggestions that improved our manuscript quality. Thank you
We revised the manuscript according to your suggestions. All revisions are clearly highlighted using the “Track Changes” function and some changes are incorporated in red text in the revised version. We provide a point-by-point specific response to your issues, with each reply being prefaced by “Response”.
We hope that this new version will meet your satisfaction.
Best regards
Point 1: This manuscript uses the MaxEnt species distribution model (SDM) with the present-day occurrences of the focal species, based on thorough field surveys (at least, I guess they were thorough; I found no information about these field surveys except their dates in L365), to model habitat suitability for that species, both now and under future climate scenarios. The authors conclude that the species is currently restricted to a few small sites in a small area of Sinai, that the total suitable habitat area is small (~70km2), that this suitable area is likely to decrease with climate change, and that other previously known populations of the species are now extinct. The focal species is critically endangered and of considerable conservation interest, making this a worthwhile study.
Response 1: Thank you for your comment. Actually, for field surveys, we added additional information. We added “using a GPS (Garmin e-Trex 20)” and “All the currently known localities with the species were visited etc.” in section 4.1. in materials and methods.
Point 2: SDMs can be very powerful tools for interpolative purposes, such as finding suitable sites for species, as done here, and indeed this is an approach that has worked well in the Sinai Protectorate for other species. When using SDMs for predicting future distributions under climate change, many assumptions are made about transferability, etc, and the results are much less reliable; even so, the approach is commonly used and may have some value if the caveats are taken properly into account. As far as I can tell, the modelling has been done quite well and many of the caveats have been taken into account. However, the explanation of the methods can improve and I have some concerns about some of the specifics of those methods (based in part on having to guess). Overall, I think the results should be reasonably robust to my concerns about the methods, and are unlikely to change qualitatively, but they may change a bit quantitatively.
Response 2: Thank you for your comment. We actually considered all of your suggestions to improve the methods and results. Please see changes in the main text in both materials and methods, and results.
Point 3: My main concern is about whether there is any circularity in the results for reduction in suitable area with climate change. I am not sure about this because the method has not been well enough explained. Do the predicted suitable sites in the future (Figure 4) include areas outside the range of the data in the models? For example, the study area includes elevations up to 2640m (L344) but there is no representation in the training data of sites above 2500m (Fig.2). Yet Fig.2 suggests that an elevation of ~2600m would be highly suitable. If such areas were excluded (as implied by L285) then the result that a loss of suitable area for the species is expected in the future is partly circular. If there is a problem here, I do not expect it to qualitatively change the finding (I would still expect reductions in the suitable habitat), but it could change it quantitatively (the predicted amount of reduction).
Response 3: We appreciate your comment. To compare the current climatically suitable areas and projected future scenarios, we repeated the calculations in ArcGIS, and we confirm the amount of reduction in the predicted suitable area. We revised the sentence (L285), and changed it as follows” high elevation (> 2000 m)”. For information, we did not exclude areas above 2600 m. In addition, we used a binary map (0: unsuitable, 1: suitable) at a threshold value of 0.60 for easy comparison, but in figure 2 the suitability is a continuous output (0.0- 1.0). All the predicted suitable sites were included in the range of the study area. The loss percent in a suitable area is the difference between the loss and gain suitable area between the future projection and the current climatically suitable area. To improve the explanation of our methods we added “then the loss or gain in suitability area was calculated given the difference between the current and future climatically suitable areas” in the last paragraph in section 4.3. in materials and methods. Thanks again.
Point 4: The procedure to reduce collinearity problems, as described in L392-98, is also problematic. The VIF basically measures the overlap of explanatory power between the focal variable and the other variables in the data. It is good for highlighting where problems are, but it is not appropriate to just remove all variables with high VIF, which is what was apparently done. A simple example illustrates the problem: if you have two variables that measure the same thing and are highly co-linear then both would have a high VIF; these variables might for example be mean temperature and maximum temperature. If those were the only two temperature variables in the data then removing them both would remove temperature completely from the analysis, which is likely to remove an important signal. Given the strong overlap in explanatory power, the correct procedure would be to remove one of the two temperature variables and then re-examine the VIF - in this case, the VIF of the other one would then become very low. The decision on which of the two to remove is best done using theoretical reasoning (which variable is the most appropriate to keep?) Do NOT remove both variables just because they both have high VIF! It looks as though the authors have made this mistake. If so, they should check whether their analysis should be redone. The good news is that I do not expect any such new analysis to alter the results very much. This is because the variables the authors did include seem to me to have been appropriate ones. Perhaps that was good luck! Or perhaps it was actually due to the authors using their judgment about which variables are appropriate to include, and not just about removing all variables with high VIF- in which case the results are fine but the authors should rewrite that part of the methods to explain what they actually did!
Response 4: We appreciate your comment. In fact, in our study, to assess collinearity, we used the variance inflation factor (VIF). As you mentioned in your comment, at first, we followed “the stepwise selection progresses”. We calculated the VIF values for all variables, then removes the variable with the highest value and repeats until all variables with VIF values below 5 were kept. We added the following part to section 4.2. in the material and methods “We performed a stepwise variable selection: first, VIF values for all variables were calculated, then the variable with the highest VIF was iteratively removed until no variables with VIF greater than the threshold (5) remains [9, 14]”.
Point 5: My other main comment is about the likely causes of local extinctions and the overall endangerment of the species. L278-80 and section 3.4 cover some issues of clear relevance, but do not mention grazing. Surely grazing is a key pressure on the species, too? My guess is that it is very sensitive to grazing (perhaps by goats?), which might explain why it is most often found on very steep cliffs! If grazing is indeed important then an important part of trying to conserve the species would presumably be to protect it from grazing as much as possible.
Response 5: Yes, the current study and previous studies displayed that grazing is considered as one threat for P. boveana. Therefore, we added “sheep and goat grazing” in section 3.2. as one of the causes of local extinctions. In addition, we added in section 3.4. “In accordance with previous studies [57–59], our field observations displayed that grazing is a key pressure on P. boveana and this might explain why this species is often found on very steep cliffs”. In addition, we added “and further it is important to protect it against grazing” in conclusions. Thank you very much for your comment.
References
[57] Mansour, H.; Jiménez, A.; Keller, B.; Nowak, M. D.; Conti, E. Development of 13 Microsatellite Markers in the Endangered Sinai Primrose (Primula boveana, Primulaceae). Appl. Plant Sci., 2013, 1 (6).
[58] Jiménez, A.; Mansour, H.; Keller, B.; Conti, E. Low Genetic Diversity and High Levels of Inbreeding in the Sinai Primrose (Primula boveana), a Species on the Brink of Extinction. Plant Syst. Evol., 2014, 300 (5), 1199–1208.
[59] Omar, K. Assessing the Conservation Status of the Sinai Primrose (Primula boveana). Middle-East J. Sci. Res., 2014, 21 (7), 1027–1036.
Point 6: Throughout the manuscript, attention is needed to the English. However, in most cases the meaning is reasonably clear.
Response 6: Ok, we revised and improved it. Thank you
Point 7: In the abstract, ‘we confirmed five extinct localities’ (L32) is not very clear. This eventually becomes somewhat clear (though not very), later in the manuscript, but it should be better explained in the abstract.
Response 7: Thank you for your comment. In the abstract, we add “we confirmed five extinct localities’ where P. boveana has been previously recorded”.
Point 8: L386 states that the elevation data were 1km2 resolution, and from these slope and aspect were calculated. Surely a 1km2 raster digital elevation model is much too coarse to calculate slope and aspect! (Thus I wondered, on reading this, why you did not use the SRTM digital elevation model, available at 90m and 30m resolution.) Yet the slope values for Table 1 suggest that the slope and aspect cannot have been calculated with 1km2-resolution elevation data. So I suspect there is a mistake in explaining the dataset and/or calculation procedure.
Response 8: In fact, we preferred to use the same resolution (1 km2) for all variables to avoid “resampling” that increase uncertainty (Alsamadisi et al. 2020). In most studies, the selection of resolution is a consequence of the availability and quality of data pertaining to the specific study area, which is typically the limiting factor in distribution studies (Elith et al. 2006). Data layers used in such studies are most commonly derived from global databases, in which 1 km2 is considered the finest resolution (Nezer et al. 2017). We used the high spatial resolution (30 arc-seconds, ~1 km at the Equator) climate datasets of WorldClim11. We chose WorldClim due to its relatively high spatial resolution, wide use and quality. Furthermore, for most mountains regions, resampling will increase bias and autocorrelation among the climate, topography and edaphic variables. In our study, from elevation raster layer (DEM, digital elevation model), we derived aspect and slope using spatial analyst tool in ArcGIS. Aspect identifies the downslope direction of the maximum rate of change in value from each cell to its neighbors while slope represents the rate of change in elevation for each digital elevation model cell. The same method was applied in several similar studies (e.g. Abdelaal et al. 2019, Mohammadi et al. 2019, Yan et al. 2020, etc.). Also, we added “In order to avoid resampling procedures, which are often a source of uncertainty [85]” in section 4.2. In addition, we added “Some ecological explanations may be missed because of the lack of a large set of variables at a higher resolution” in the second paragraph in section 3.1. in discussion. Thank you.
References
[9] Abdelaal, M., Fois, M., Fenu, G., & Bacchetta, G. (2019). Using MaxEnt modeling to predict the potential distribution of the endemic plant Rosa arabica Crép. in Egypt. Ecological informatics, 50, 68-75.
Alsamadisi, A. G., Tran, L. T., & PapeÅŸ, M. (2020). Employing inferences across scales: Integrating spatial data with different resolutions to enhance Maxent models. Ecological Modelling, 415, 108857.
[39] Elith, J., H. Graham, C., P. Anderson, R., Dudík, M., Ferrier, S., Guisan, A., ... & Li, J. (2006). Novel methods improve prediction of species’ distributions from occurrence data. Ecography, 29(2), 129-151.
Mohammadi, S., Ebrahimi, E., Moghadam, M. S., & Bosso, L. (2019). Modelling current and future potential distributions of two desert jerboas under climate change in Iran. Ecological Informatics, 52, 7-13.
[33] Nezer, O., Bar-David, S., Gueta, T., & Carmel, Y. (2017). High-resolution species-distribution model based on systematic sampling and indirect observations. Biodiversity and Conservation, 26(2), 421-437.
Yan, H., He, J., Zhao, Y., Zhang, L., Zhu, C., & Wu, D. (2020). Gentiana macrophylla response to climate change and vulnerability evaluation in China. Global Ecology and Conservation, 22, e00948.
Point 9: Figure 1 and Table S1: given the strong correlation between size and density, why does Figure 1 show them as apparently uncorrelated? This is not necessarily an error, but requires some explanation.
Response 9: We agree completely with your comment. We repeated the analyses (CCorA and Pearson correlation) and gave the same result. At first, as you know CCorA is a multivariate analysis while Pearson is bivariate. Along CCorA, the incoherence between size index and density may be attributed to at least one variable was not normally distributed. We also found the size index was closely related to CCorA axis 1 (-0.605) while density was 0.282. In addition, the size index may also be affected by another variable which led to alter its correlation with the density. Density and size index may increase simultaneously with the 2nd axis (mainly according to aridity index, Bio6, Bio7, Bio15, aspect) while they differ according to the 1st axis (mainly according to clay and slope). Thank you
Point 10: In Table 2, you need to explain in the caption what the various models are (e.g. E. model = edaphic only), as done in Figure 3. Readability is seriously reduced if the reader has to search the manuscript to interpret this table.
Response 10: Thank you for your suggestion. Now, in Table 2, we replaced all abbreviations (acronyms) for model-type name by the complete type-name. For example, “C. model” is replaced by “Climate-only model”, etc.
Point 11: Similarly, in the main text, it would be better not to use these abbreviations. It would be quite succinct enough, and much more readable, to refer to the models as, for example, ‘edaphic-only model’, ‘climate-topography model’, etc.
Response 11: We replaced all abbreviations for all the model-type names in the main text by the complete type-name. For example, “C. model” is replaced by “Climate-only model”, etc. Thank you.
Point 12: In Figure 2, it would be better for the vertical axes to all have the same scale.
Response 12: Actually, Figure 2 with its scale is automatically released in MaxEnt outputs. However, we improved the scale (except for organic carbon: 0.0-0.6 and Bio14: 0.0-0.8) and also the scale become homogenous in all variables with one decimal number. Please see the changes in the new version of Figure 2. Thank you.
Point 13: In reporting the results illustrated in Fig.2, the authors should bear in mind that the curve does not peak in most cases, but instead has its maximum value at one end of the data range. Therefore, it is not appropriate to make statements such as ‘suitable elevation range was between 1600 m up to 2500 m. a.s.l.’. In this case it would be much more appropriate to say that the most suitable elevations were the highest ones sampled, or at least to point out that no reduction in suitability was found at the highest elevations. Similar considerations apply to most of the other statements about the findings shown in Fig.2.
Response 13: In fact, we agree with your comment and explanation. Now, all statements about the findings that are shown in Figure 2 are revised and changed in the results and discussion. Please see changes in section 2.2. in the results (L167-L177) and in discussion (L291-L292). Thank you.
Point 14: It is also strange that the statements about ‘suitable range’ encompass almost all non-zero suitability values. Taking the example of elevation again, 1600m is specified as ‘suitable’ but the suitability shown on Fig.2 for this elevation is only 0.1, which contradicts the suitability statements later in the text (e.g. L181), and Fig.3H, where ‘suitable’ is defined as >0.60. Again, similar considerations also apply to the other statements in the text about the results shown in Fig.2.
Response 14: Thank you for your comment, please see the above response. We revised the data in the results to be compatible with the suitability threshold (>0.60).
Point 15: Section 3.1 in the discussion includes a lot of text about the soil variables, when really the message seems rather simple: that edaphic factors alone are not enough to model habitat suitability for this species, but they can help fine-tune a model that captures the stronger influences of climate and topography. This lengthy discussion of soil influences seems quite pointless because surely no-one would think it sensible to ONLY include soil variables in SDMs!
Response 15: Now “section 3.1 in the discussion” is reduced and improved by your suggestions. Please see the changes in the main manuscript. Thank you.
Point 16: L436 ‘The selection of the best overfitting model’. Very strange statement; surely you do not mean this! Should it be ‘The selection of the best-fitting model’?
Response 16: Ok, in L436, we replaced “the best overfitting model” by “the best-fitting model”. Thank you.
Point 17: I wonder whether the sites referred to as ‘Wadi Shag Mousa’ (L216, 469) and ‘Wadi Shaq Mousa’ (L109, Table 1) are actually the same site?
Response 17: Actually both names are referred to the same site, however, we preferred to use the most common name “Wadi Shaq Mousa”. Now all “Wadi Shag Mousa” is replaced by “Wadi Shaq Mousa” in all the manuscript (L216 & L469). Thank you.
Reviewer 2 Report
It is a good work, but it lacks a general bibliographic review, especially in relation to other works of MaxEnt, on other works of Primula sp. with similar characteristics, etc. Some suggestions:
- http://dx.doi.org/10.1080/11263504.2014.976289
- https://doi.org/10.1016/j.gecco.2019.e00563
They need to adapt the text in general to the order of the sections of the magazine. That is to say, as the chapter on material and methods comes after the chapter on results, you must try to ensure that the acronyms, the names of the models you run, the names of the Bioclim variables, etc. are explained the first time you refer to them, otherwise the text will not be understood.
If you are going to use Bioclim variables, it is necessary to express it, even in the introduction, so that it is easily understood.
You must reinforce the section on the census, you say you extrapolate, but how? where?
They have to reinforce the conclusions, they are very poor and it does not seem that they summarize all the work

Author Response
Dear Reviewer,
We really appreciate your constructive valuable comments and suggestions that improved our manuscript quality. Thank you
We revised the manuscript according to your´ suggestions. All revisions are clearly highlighted using the “Track Changes” function and some changes are incorporated in red text in the revised version. We provide a point-by-point specific response to your issues, with each reply being prefaced by “Response”.
We hope that this new version will meet your satisfaction.
Best regards
Comment: It is a good work, but it lacks a general bibliographic review, especially in relation to other works of MaxEnt, on other works of Primula sp. with similar characteristics, etc. Some suggestions:
- http://dx.doi.org/10.1080/11263504.2014.976289
- https://doi.org/10.1016/j.gecco.2019.e00563
They need to adapt the text in general to the order of the sections of the magazine. That is to say, as the chapter on material and methods comes after the chapter on results, you must try to ensure that the acronyms, the names of the models you run, the names of the Bioclim variables, etc. are explained the first time you refer to them, otherwise the text will not be understood. If you are going to use Bioclim variables, it is necessary to express it, even in the introduction, so that it is easily understood.
You must reinforce the section on the census, you say you extrapolate, but how? where?
They have to reinforce the conclusions, they are very poor and it does not seem that they summarize all the work
Response: We added all the suggested references, for example, Alfaro-Saíz et al. (2019) [80] who studied Primula pedemontana; Alfaro-Saíz et al. (2015) [20] who used species distribution models for searching new populations of threatened flora. In addition, we add all suggested references related to MaxEnt (e.g. Elith et al. 2011[23], Fois et al. 2015 [4]). We adapted the text according to the order of chapters of the Journal. We added the full names of all text when mentioned for the first time. We changed the acronyms of the models, for example the “C. model” becomes “climate-only model”, etc. also, all bioclimatic variables when mentioned for the first time, the complete name was mentioned, for example, Bio6 becomes minimum temperature of coldest month and so on. For bioclimatic variables, we added for introduction the following sentence “As the climate is one of the main driving for species distribution, we used the bioclimate variables of the WorldClim database due to its high resolution (~1 km), broad use and quality [33]”. We reinforced the section on the census and changed the sentences, please see paragraph 3 in section 4. 1 in materials and methods. Finally, in conclusion, we improved it to summarize all of the work. Please see “conclusion”. Thank you.
#Response for comments included in the “PDF file review”
Point 1: Line 48, bibliographic references needed
Response 1: Thank you for your comment. Ok, we added “Fois et al. 2015 [4]”.
Point 2: Line 64, bibliographic references needed.
Response 2: Thank you for your comment. Actually, references are included in the end of the sentence as “[4,13,14].
Point 3: ELITH references
Response 3: Ok, we added Elith et al. 2011 [23]. Thank you.
Point 4: Needless to say, it is the acronym IUCN and cites the IUCN.
Response 4: We appreciate your comment. We added “International Union for Conservation of Nature” and already it referenced in [34].
Point 5: Line 119, m.
Response 5: Ok, we deleted a.s.l. and kept m. Thank you
Point 6: Line 125/126, Is this a direct observation of the authors?
Response 6: In fact, this information was collected from “local Bedouin observations” so we added “(local Bedouin observations”. Thank you.
Point 7: Line 145, It's the first time it's been named. It should be explained that it is
Response 7: Thank you for your comment. Now we replaced all abbreviations (acronyms) of model-type name by the complete type-name. For example, “C. model” is replaced by “Climate-only model”, etc.
Point 8: Line 151, It's the first time it's been named. It should be explained that it is
Response 8: Ok, we replaced all acronyms by the complete names. We added “precipitation of driest month (Bio14), precipitation seasonality (Bio15) and minimum temperature of coldest month (Bio6)”, etc. Thank you.
Point 9: Line 221, Explain what it is
Response 9: Ok, we added “representative concentration pathways scenarios (minimum RCP2.6 and maximum RCP8.5)”. Thank you
Point 10: Lines 366-70, Further explain this paragraph and discuss with the reviewers in research papers where this is used.
Response 10: Thank you for your comment. We improved this paragraph as follows “. All the currently known localities with the species were visited and ten permanent plots of 5 × 5 m each were centrally positioned to measure the following parameters: population size (estimated number of individuals in each subpopulation), plant density (number of individuals per plot area), size index (average of heights and diameters (cm) of three randomly individuals per plot), plant vigor (the ratio between plant size and number of leaves), and cover percent (visually estimated)”. We added the relative reference, Shaltout et al. (2015) [82].
Point 11: Line 394, Bibliographic citations?
Response 11: Ok we added these references [4,9,74]. Thank you.
Round 2
Reviewer 1 Report
In the revision, you have addressed most of the main points and the manuscript is close to being ready, in my opinion.
Evaluating the revisions was hindered by the line numbers in your response letter not matching those in the revised manuscript. When they do not match, it adds frustration and time taken to evaluate the revisions. For example, in your response 3 you talked about a change in L285, quoting the new text, but this exact text does not appear anywhere in the manuscript! It appears to refer to L306, but to determine this I had to open the pdf of the first submission to the journal and cross-check.
With respect to response 3, it is good to know that there is no circularity problem, but please make this clear in the manuscript itself (not only in the response letter!). The key part of your response is clarifying that environmental space not included in the training data but present in the study area (such as sites >2600m elevation) was not excluded from the predicted future distributions. This needs explaining in the manuscript, including making clear how you determined what suitability values were used in such cases (e.g. for elevations >2500m, did you assume the same suitability as for 2500m?).
Multicollinearity: this is now better explained, and it is reassuring to know that you did not make the error of removing whole suites of variables in one step. However, automated procedures such as the stepwise one used here, while having the advantage of being easily repeatable, emphasise statistical fitting rather than biological sense. As I mentioned before, it turns out that the set of variables that this procedure happened to select in this case do seem sensible ones, so I do not think reanalysis is needed. However, I do think the authors should comment on the appropriateness of the set of variables used in the modelling – explaining why it is appropriate to use these particular variables to model the distribution of the species.
The slope and aspect data still need better explanation, I think. The data seem to be OK as far as I can tell, but not the explanation of how they were derived. The methods currently suggest that a single elevation value for each ~1km2 grid cell was used, and then slope and aspect data were calculated from these very coarse elevation data. I cannot see how using a 1km2-resolution digital elevation model could give slope values of 90 degrees (e.g. 6 of the 10 sites in Table 1) – what values of elevation that are 1km apart can give a value of 90 degrees for slope? It seems to be a mathematical impossibility. Values like that must surely have come from fine-resolution locality data. Similarly, the slope values up to 75deg shown in Fig.2 seem extremely unlikely to have been calculated from a 1km-resolution DEM (a slope of 75deg for points 1km apart represents an elevational difference of ~3700m, which is more than the entire elevational range of the study area). What am I missing, here?
What is meant by the following in L498? ‘This is especially true when models are constructed from an initial number of initial locations.’ This new text is not clear at all.
Finally, there are still lots of minor issues with the English. I suppose these can be picked up in copy-editing.
Author Response
Response to Reviewer 1 Comments (2nd Round)
Dear Reviewer,
We really appreciate your constructive valuable comments and suggestions that improved our manuscript quality. Thank you
We revised the manuscript according to your suggestions. All revisions are clearly highlighted using the “Track Changes” function and some changes are incorporated in red text in the revised version. We provide a point-by-point specific response to your issues, with each reply being prefaced by “Response”.
Best regards
Point 1: In the revision, you have addressed most of the main points and the manuscript is close to being ready, in my opinion.
Response 1: Thank you very much. We hope this new version will meet your satisfaction.
Point 2: Evaluating the revisions was hindered by the line numbers in your response letter not matching those in the revised manuscript. When they do not match, it adds frustration and time taken to evaluate the revisions. For example, in your response 3 you talked about a change in L285, quoting the new text, but this exact text does not appear anywhere in the manuscript! It appears to refer to L306, but to determine this I had to open the pdf of the first submission to the journal and cross-check.
Response 2: We apologize for the unintended error. This happened due to a shift in the line numbers during “track changes” in the word”. Please accept our apologies.
Point 3: With respect to response 3, it is good to know that there is no circularity problem, but please make this clear in the manuscript itself (not only in the response letter!). The key part of your response is clarifying that environmental space not included in the training data but present in the study area (such as sites >2600m elevation) was not excluded from the predicted future distributions. This needs explaining in the manuscript, including making clear how you determined what suitability values were used in such cases (e.g. for elevations >2500m, did you assume the same suitability as for 2500m?).
Response 3: We added “All the predicted suitable sites in the future were included in the range of the study area. In detail, sites of >2500 m elevation that were not included in the training data but present in the study area were not excluded from the predicted future distributions and considered with a suitability value of 0.90” in section 2.5. (Lines 254-257). Also, we added “Finally, in future projections, uncertainty may increase, especially when the environmental predictors (such as elevation in our study) need to be extrapolated outside the range of the training data of the species’ response, therefore physiological, biological, and distributional attributes of the species should be considered” in section 3.4, Lines 374-378. Thank you
Point 4: Multicollinearity: this is now better explained, and it is reassuring to know that you did not make the error of removing whole suites of variables in one step. However, automated procedures such as the stepwise one used here, while having the advantage of being easily repeatable, emphasise statistical fitting rather than biological sense. As I mentioned before, it turns out that the set of variables that this procedure happened to select in this case do seem sensible ones, so I do not think reanalysis is needed. However, I do think the authors should comment on the appropriateness of the set of variables used in the modelling – explaining why it is appropriate to use these particular variables to model the distribution of the species.
Response 4: We added “Similar to the other mountain species of St. Catherine that are sensitive to high altitude, temperature, rainfall, and soil features [75, 78], we assumed that the selected variables were appropriate for defining the ecology and spatial distribution of P. boveana” In section 4.2., Lines 448-451 in the new revised version of the manuscript. Thank you.
Point 5: The slope and aspect data still need better explanation, I think. The data seem to be OK as far as I can tell, but not the explanation of how they were derived. The methods currently suggest that a single elevation value for each ~1km2 grid cell was used, and then slope and aspect data were calculated from these very coarse elevation data. I cannot see how using a 1km2-resolution digital elevation model could give slope values of 90 degrees (e.g. 6 of the 10 sites in Table 1) – what values of elevation that are 1km apart can give a value of 90 degrees for slope? It seems to be a mathematical impossibility. Values like that must surely have come from fine-resolution locality data. Similarly, the slope values up to 75deg shown in Fig.2 seem extremely unlikely to have been calculated from a 1km-resolution DEM (a slope of 75deg for points 1km apart represents an elevational difference of ~3700m, which is more than the entire elevational range of the study area). What am I missing, here?
Response 5: We apology for being unclear. All these variables were used at 1 km resolution because this was the best available resolution for all data. Indeed, different resolutions are not admitted by Maxent. So, what we did was extracting slope and aspect from a high-resolution digital elevation model (90 m) and then aggregating them using the median approach, as suggested in Amatulli et al. (2018). Please, see the implementation of this aspect in section 4.2. in methods (L 433-436).
Reference
[86] Amatulli, G.; Domisch, S.; Tuanmu, M. N.; Parmentier, B.; Ranipeta, A.; Malczyk, J.; Jetz, W. Data Descriptor: A Suite of Global, Cross-Scale Topographic Variables for Environmental and Biodiversity Modeling. Sci. Data, 2018, 5. https://doi.org/10.1038/sdata.2018.40.
Point 6: What is meant by the following in L498? This is especially true when models are constructed from an initial number of initial locations. This new text is not clear at all.
Response 6: We improved it as follows “This is especially true when models are constructed from original presence records of the native locations for the species”. Please, see changes in Lines 520-521 in conclusions. Thank you.
Point 7: Finally, there are still lots of minor issues with the English. I suppose these can be picked up in copy-editing.
Response 7: Thank you for your comment. We improved the English of the manuscript. We are thankful to Martin Barry for giving linguistic advice for the manuscript.
